# Rapid and Ultrasensitive Detection of *H. aduncum* via the RPA-CRISPR/Cas12a Platform

**DOI:** 10.3390/molecules29204789

**Published:** 2024-10-10

**Authors:** Xiaoming Wang, Xiang Chen, Ting Xu, Xingsheng Jin, Junfang Jiang, Feng Guan

**Affiliations:** 1College of Life Sciences, China Jiliang University, Hangzhou 310018, China; wxm19980319@163.com (X.W.); xxxxudsr@163.com (T.X.); 2Zhejiang Museum of Natural History, Hangzhou 310018, China; jinxs@zmnh.com; 3Zhoushan Institute for Food and Drug Inspection and Testing, Zhoushan 316021, China; 13615805498@163.com; 4Institute of Animal and Veterinary Science, Academy of Zhejiang Agriculture Science, Hangzhou 310021, China

**Keywords:** *H. aduncum*, recombinase polymerase amplification (RPA), CRISPR/Cas12a, visual rapid detection

## Abstract

*Hysterothylacium aduncum* is one of six pathogens responsible for human anisakiasis. Infection with *H. aduncum* can cause acute abdominal symptoms and allergic reactions and is prone to misdiagnosis in clinical practice. This study aims to enhance the efficiency and accuracy of detecting *H. aduncum* in food ingredients. We targeted the internal transcribed spacer 1 (ITS 1) regions of *Anisakis* to develop a visual screening method for detecting *H. aduncum* using recombinase polymerase amplification (RPA) combined with the CRISPR/Cas12a system. By comparing the ITS 1 region sequences of eight nematode species, we designed specific primers and CRISPR RNA (crRNA). The specificity of RPA primers was screened and evaluated, and the CRISPR system was optimized. We assessed its specificity and sensitivity and performed testing on commercial samples. The results indicated that the alternative primer ADU 1 was the most effective. The final optimized concentrations were 250 nM for Cas12a, 500 nM for crRNA, and 500 nM for ssDNA. The complete test procedure was achievable within 45 min at 37 °C, with a limit of detection (LOD) of 1.27 pg/μL. The amplified product could be directly observed using a fluorescence microscope or ultraviolet lamp. Detection results for 15 *Anisakis* samples were entirely consistent with those obtained via Sanger sequencing, demonstrating the higher efficacy of this method for detecting and identifying *H. aduncum*. This visual detection method, characterized by simple operation, visual results, high sensitivity, and specificity, meets the requirements for food safety testing and enhances monitoring efficiency.

## 1. Introduction

*Hysterothylacium aduncum* is a member of the phylum Nematoda, class Phasmidia, order Spirurida, family *Anisakidae*, subfamily *Ascaridina*, and genus *Hysterothylacium*. It infects not only marine bony fish but also various fish species in freshwater and at river–sea intersections. Reports indicate infections in over 220 fish species across 70 families and 22 orders globally [1,2]. In some surveys, infection rates have reached 80% or even 100% [3,4]. Along with *Anisakis simplex (s. s.)*, *A. pegreffii*, *A. typica*, *Pseudoterranova decipiens*, and *Contracaecum osculatum*, *H. aduncum* is a major pathogen of anisakiasis. Infection with these parasites can lead to acute abdominal symptoms and allergic reactions, and is frequently misdiagnosed clinically [5]. *H. aduncum*, found in food sources, is a primary cause of this disease, highlighting the importance of enhancing food source monitoring to prevent and control the disease.

Identifying *H. aduncum* relies mainly on morphological and molecular biological techniques. Morphological identification involves observing species’ characteristics under a microscope. However, identifying species based on morphology is challenging, particularly for larvae in the egg stage or during early infection stages in fish and humans [5,6,7]. Molecular identification methods, primarily PCR-based, are commonly used to identify *Anisakis* and *H. aduncum* [8,9]. While these methods are highly sensitive and specific, they require sophisticated equipment and specialized skills, making them unsuitable for rapid on-site detection [10]. Effective parasite surveys and nematode identification require distinguishing *H. aduncum* from other *Anisakis* species to enhance prevention and control measures and develop specific strategies. Current molecular detection techniques for *H. aduncum* include morphology combined with DNA barcoding [11], qPCR [12], loop-mediated isothermal amplification with lateral flow test dipstick (LAMP-LFD) [13], and PCR-RFLP. Despite their efficacy, these methods are inadequate for the rapid detection needed in food and clinical settings. Thus, developing a faster, more accurate, and simpler detection method for *H. aduncum* is essential to meet the growing demands for efficient detection.

Recombinase polymerase amplification (RPA) is a relatively novel isothermal amplification technique based on recombination proteins, which offers advantages such as rapidity, accuracy, and the ability to perform amplification at room temperature, making it ideal for on-site detection [14,15]. It has been extensively used for detecting and identifying pathogens including bacteria [16], fungi [17], viruses [18], and parasites [19]. RPA can amplify the target gene within 10–30 min, fulfilling the requirement for rapid detection. The optimal reaction temperature for RPA is around 37 °C, which is easy to control but can be prone to non-specific fragments and false-positive results [20]. However, combining RPA with the CRISPR/Cas system enhances specificity and sensitivity. The Cas12a protein, active in cleaving single-stranded DNA (ssDNA) within a collateral cleavage system [21], also facilitates visualization [22,23]. This combination provides valuable insights for the molecular detection of *H. aduncum*.

This study utilizes the ITS 1 sequences of *H. aduncum* as the target gene to develop a visual fluorescence detection method based on the RPA-CRISPR/Cas12a platform, offering technical support for detecting and identifying *H. aduncum* in seafood and processed products.

## 2. Results and Discussion

### 2.1. Nematode DNA Quality and RPA Primer Selection

Twenty and fifteen DNA samples, yielding an average concentration of 9.21 ± 0.16 ng/μL and an A260/A280 purity ratio ranging from 1.8 to 2.2 (average 1.89 ± 0.42), indicating that the extracted DNA was suitable for further analysis. Among the three designed RPA primer sets, the ADU1 primer set demonstrated the best amplification efficiency, producing no non-specific fragments, and the amplified target fragment matched the expected length (see Figure 1). The positions of the selected ADU1 primer set and the crRNA in the ITS 1 sequences of *H. aduncum* are shown in Figure 2.

*H. aduncum* is one of the six pathogens known to cause anisakiasis. It primarily parasitizes marine bony fish, such as anglerfish and cod, and is also frequently found in various freshwater and river–sea interface fish, with a prevalence reaching up to 60% [2]. A literature review revealed that only a few clinical cases caused by *H. aduncum* have been identified [24,25]. This may be because most clinical cases rely solely on morphological identification, with only about 7% of reported anisakiasis cases including species identification using molecular techniques [26]. Morphological identification of nematodes is significantly limited by the high degree of similarity among *Anisakis* species. Moreover, nematodes ingested with food are often fragmented by the digestive tract, losing their species-specific features but still causing anisakiasis, leading to potential misclassification [6,27,28]. Accurate pathogen identification is crucial for economic and clinical reasons, as it helps identify the infection source, prevent transmission, and enhance understanding of pathogen mechanisms. It also facilitates timely preventive and therapeutic measures. Currently, many DNA-based detection methods are widely used in parasite detection fields, this assay was also developed based on this idea.

### 2.2. Specificity of the RPA-CRISPR/Cas12a Assay

The RPA-CRISPR/Cas12a specificity assay results indicated that significant fluorescence signals were observed only in tube 1 containing *H. aduncum* DNA, as measured by the fluorescence quantitative PCR instrument (Figure 3A) and under UV light (Figure 3B). All other samples, including *A. simplex* (*s. s.*), *A. typica*, *H. sinense*, *Contracaecum* spp., mixed DNA from small yellow croaker and hairtail, and ddH_2_O, were negative, consistent with the RPA results (Figure 3C). These results demonstrated that the RPA primer set and the CRISPR/Cas12a system had high specificity with no cross-reactivity to DNA from other closely related species or fish hosts.

The detection of nematode pathogens in *Anisakis* primarily focuses on *A. simplex* (*s. s.*), *A. pegreffii*, and *A. typica* [9,29], with relatively few reports on the species identification of *H. aduncum*. Among existing reports, only Qiao Yan et al. [13] developed a detection system for *H. aduncum* using LAMP-LFD, which can be completed in 30–40 min with a sensitivity of 1.4 × 10^2^ copies/μL. Conventional techniques, such as PCR, PCR-RFLP, and even the LAMP method, are limited by the necessity for laboratory equipment [30,31]. These conventional PCR-based methods are time-consuming, typically requiring 2.5–4 h, and do not meet the rapid detection needs of food supervision. Additionally, the LAMP method faces challenges with primer design [32]. Common RPA methods also experience issues such as aerosol-induced false positives and low amplification efficiency, and product observation still requires electrophoresis and other complex processes, which hinders their application in rapid detection [22,33]. The CRISPR/Cas12a system, with an optimal reaction temperature of 37 °C, can be used in combination with RPA to enhance detection sensitivity by amplifying the signal. This RPA-CRISPR/Cas12a combination reduces non-specific amplification by RPA and improves reaction sensitivity, addressing the demand for rapid detection and has been applied to *Anisakis* identification [9,23].

### 2.3. Optimization of Cas12a/crRNA Concentration Ratio

Optimization of the Cas12a/crRNA concentration ratio was performed to determine the ideal reaction system based on the fluorescence intensity of the RPA reaction products. The optimized results are shown in Figure 4. As the Cas12a/crRNA concentration ratio was gradually increased from 1:1, the fluorescence intensity of the reaction products peaked at a ratio of 1:2 (61,011 ± 3417), significantly higher than at 1:1 (38,754 ± 4006, *p* < 0.01). Intensity then decreased with further increases in the concentration ratio. The difference in fluorescence between ratios of 1:2 and 1:3 was not significant (*p* < 0.05), which was also reflected in the lack of visible difference in color under UV light (Figure 4B). To balance reagent costs with practical detection requirements, a Cas12a/crRNA ratio of 1:2 was chosen as the final optimization, corresponding to final concentrations of 250 nM for Cas12a and 500 nM for crRNA. The combination of Cas12a and crRNA has the activity of cutting single-stranded DNA molecules, which is the principle of this method to achieve good visual detection results, so the effect of the ratio of Cas12a/crRNA concentration was explored in the system in the experimental results. The results showed that the system had the best detection effect when the ratio of Cas12a/crRNA concentration was 1:2.

### 2.4. Optimization of ssDNA Concentration

A comparative analysis of the fluorescence intensity of the RPA reaction system with varying ssDNA concentrations revealed an increasing fluorescence intensity with higher ssDNA concentrations (Figure 5A). Increasing the ssDNA concentration from 300 nM to 400 nM significantly enhanced the fluorescence intensity of the reaction products (29,631 ± 957 vs. 48,750 ± 4428, *p* < 0.01). Further increasing the concentration to 500 nM resulted in greater fluorescence intensity (54,063 ± 3885). Although this increase was not as significant compared to the 400 nM concentration (*p* > 0.05), it provided more favorable visual results under UV light (Figure 5B). Consequently, a ssDNA concentration of 500 nM was selected as the optimal condition for the system. Optimization of the ssDNA concentration is an important part of the CRISPR system for visualizing the results, which has a significant impact on the sensitivity and specificity of the results. From the present results, it can be observed that the optimized concentration fully meets the requirements of the assay in terms of both specificity and sensitivity.

### 2.5. Sensitivity of the RPA-CRISPR/Cas12a

The sensitivity of the RPA-CRISPR/Cas12a system was assessed using serial dilutions of *H. aduncum* DNA. As the DNA concentration decreased, the amount of RPA amplification products correspondingly decreased (Figure 6A). At a dilution of 1.27 pg/μL, no visible electrophoretic bands were observed, while the total amount of DNA in the RPA reaction system was 6.35 pg. In contrast, when the DNA concentration was 1.27 pg/μL, there was a notable discrepancy in fluorescence intensity compared to that of the 127 fg/μL dilution (*p* < 0.01) (Figure 6B). Weak fluorescence was observed in tube 5 under UV light (Figure 6C), and this fluorescence signal was discernible to the unaided eye. The electrophoresis results and fluorescence intensity indicated that the limit of detection (LOD) of the RPA-CRISPR/Cas12a system was 1.27 pg/μL. This sensitivity was ten times higher than that of the standard RPA reaction, which had an LOD of 12.7 pg/μL.

The CRISPR/Cas system is a nucleic acid assay derived from recent CRISPR gene editing technology. It is a reliable tool with high specificity and sensitivity for nucleic acid detection. The Cas12a used in this system belongs to the second major class V type V-A subtype protein with nucleic acid endonuclease activity, which allows for the detection of double-stranded DNA and enhances detection sensitivity by amplifying the secondary signal [21]. In this study, ITS gene sequences, which are highly variable between different species of *H. aduncum* [34], were used as the target to develop an RPA-CRISPR/Cas12a assay for rapid identification of parasitic nematodes in fish and for clinical identification of suspected cases. The RPA-CRISPR/Cas12a reaction system was optimized at 37 °C, suitable for rapid on-site detection, and specifically detects *H. aduncum*. Results can be observed directly in approximately 40 min using a portable fluorometer or UV lamp, offering ease of use along with high sensitivity, specificity, and result visualization. Optimization of the reaction system components revealed that the best results were achieved with ssDNA at a concentration of 500 nM and a Cas12a to crRNA ratio of 1:2. Notably, the study indicated that the fluorescence intensity signal obtained with the fluorescence quantitative PCR instrument increased with ssDNA concentration, peaking at 500 nM. Zhao et al. [5] reported that fluorescence intensity peaked at 450 nM ssDNA concentration, with only a minor decrease at 600 nM. In contrast, Wang X et al. [33] reported that fluorescence intensity increased with higher ssDNA concentrations. For ssDNA concentrations of 400 nM and 500 nM, there were no significant differences in fluorescence signal intensity, with both concentrations meeting detection requirements. However, excessively high ssDNA concentrations can lead to increased costs and may affect detection specificity, potentially resulting in false positives. Therefore, in this study, the ssDNA concentration suitable for naked-eye observation was optimized to 500 nM for the final reaction system. Additionally, the RPA-CRISPR/Cas12a system demonstrated a minimum DNA detection concentration of 1.27 pg/μL, which was superior to the RPA system’s sensitivity of 12.7 pg/μL. This sensitivity was comparable to that of the RPA-LFD for *Anisakis* developed by our group [9] and exceeded the sensitivity of multiplex PCR reported by Paoletti M et al. [35] and the qPCR method reported by Godínez-González et al. [36]. Furthermore, testing commercial samples yielded results consistent with PCR sequencing, demonstrating strong accuracy and meeting market supervision and testing demands.

### 2.6. Commercial Sample Testing Using RPA-CRISPR/Cas12a

The optimized RPA-CRISPR/Cas12a assay was applied to identify 15 nematode samples. The identification was based on the fluorescence intensity of the reaction system. Results revealed that samples numbered 10, 11, and 13 were identified as *H. aduncum* (Figure 7). This was consistent with the PCR sequencing results and NCBI database Blast analysis. The remaining samples were identified as follows: 10 samples as *A. pegreffii* and 2 samples as *H. sinense*. These findings demonstrate that the developed RPA-CRISPR/Cas12a assay possesses excellent species specificity and is effective for the identification of nematodes. Despite the advantages gained with CRISPR/Cas12a, the method still presents several challenges. These include the high cost of raw reagents and the associated high cost of the method itself, the complexity of the design of the crRNA, the higher specific sequence requirements, the potential for off-target effects, and the difficulty of detecting multiple samples, all these factors will increase the complexity and cost of this assay. It is reasonable to posit that these issues will be resolved with the advancement of technology, thereby rendering the technology a convenient and widely available means of detection.

## 3. Materials and Methods

### 3.1. Sources of Experimental Samples

A total of 20 specimens from five species of *Anisakidae* nematodes (*H. aduncum*, *A. simplex* (*s. s.*), *A. typica*, *H. sinense*, and *Contracaecum* spp.) were obtained from laboratory-identified and stored samples [9]. Genomic DNA was extracted using an Animal Tissue Kit (Hangzhou Xinjing Biotechnology Co., Ltd., Hangzhou, China), then extracted and analyzed using Nanodrop 2000 (Thermo Fisher Scientific, Waltham, MA, USA) and stored at 4 °C for later use.

### 3.2. Main Reagents and Instruments

The LfCas12a protein, 10× Loading Buffer, DEPC-treated water, and RNase Inhibitor were purchased from Sangon Biotech (Shanghai) Co., Ltd. (Shanghai, China). The RPA detection kit was obtained from Ampure Future (Changzhou) Biotechnology Co., Ltd. (Changzhou, China). The Tellon Real-Time Fluorescent Quantitative PCR Instrument (Gentier 48E) was purchased from Xi’an Tellon Science and Technology Co., Ltd. (Xi’an, China), and the Shanghai Qinxiang Gel Imaging System (Genosens 2100) was acquired from Shanghai Qinxiang Scientific Instrument Co., Ltd. (Shanghai, China).

### 3.3. RPA Primer Design and Amplification Test

The ITS 1 sequences of *H. aduncum* were selected as the target gene for developing an RPA-CRISPR/Cas12a assay. Additionally, ITS 1 sequences of eight nematode species, including *H. aduncum* (No. KY018601.1), *A. simplex* (*s. s.*) (No. MT516319.1), *A. typica* (No. FJ161072.1), *A. physeteris* (No. KY826440.1), *A. brevispiculata* (No. KY352231.1), *Contracaecum* (No. KF990496.1), Gnathostoma (No. JN408329.1), and Raphidascaris (No. MW371020.1), were downloaded and compared to identify specific sequences. Using Clone Manager 9 software, three sets of alternative RPA primers were designed and subsequently optimized. The crRNA was designed using the crRNA design website of EZassay Biotechnology (https://ezassay.com/rna/cont/, accessed on 3 January 2024). The 5′ end of the ssDNA was modified with FAM, and the 3′ end was modified with BHQ-1. The primers and ssDNA were designed with Clone Manager 9 software and synthesized by Hangzhou Tsingke Biotechnology Co., Ltd. (Hangzhou, China). The sequences are listed in Table 1.

The RPA reaction tests were conducted using a commercial Recombinase Polymerase-based Amplification Kit (Haimps Future, Changzhou Biotechnology Co., Ltd., Changzhou, China). The reaction system was 50 μL, consisting of 2 μL of each primer (10 μM), 29.4 μL of A Buffer, 5 μL of DNA template, and 9.1 μL of ddH_2_O. After a 30-min reaction at 37 °C, 2.5 μL of B Buffer (magnesium acetate, 21 mM) was added. The product was mixed with an equal volume of phenol-chloroform (25:24:1) to remove proteins. Following centrifugation at 12,000 rpm for 5 min, the supernatant was analyzed using electrophoresis on a 2% agarose gel to compare and select the best amplification among the three alternative primer sets.

### 3.4. RPA-CRISPR/Cas12 Assay and Its Specificity

The 20 μL CRISPR/Cas12a detection system comprised 0.5 μL Cas12a protein (10 μM), 2 μL 10× buffer, 1 μL crRNA (10 μM), 0.5 μL RNase inhibitor, 3 μL RPA product, 1 μL ssDNA (10 μM), and DEPC-treated water to a final volume of 20 μL. The reaction was performed using a fluorescence quantitative PCR instrument with the following program: 30 cycles at 37 °C, each cycle lasting 30 s, with fluorescence signals collected every 30 s. After the reaction, the centrifuge tube was placed under a UV lamp in a gel imaging system for observation and photography. All experiments were repeated independently at least three times.

To determine the specificity of the CRISPR/Cas12a reaction system, *H. aduncum* was used as the positive control. Four other nematode species (*A. simplex* (*s. s.*), *A. typica*, *H. sinense*, and *Contracaecum* spp.) served as negative controls. Additionally, DNA from two fish species, small yellow croaker and hairtail, was also used as controls, with their DNA mixtures and ddH_2_O serving as negative samples. These samples were incubated at 37 °C for 45 min for RPA-CRISPR/Cas12a amplification and specificity assays, with all tests independently repeated three times.

### 3.5. Optimization for the Concentration Ratio between Cas12a and CrRNA

To optimize the specificity of the CRISPR/Cas12a reaction system, the concentration of Cas12a was kept at 250 nM, while the concentrations of crRNA were varied from 250 nM to 1.25 μM, including 500 nM, 750 nM, 1 μM, and 1.25 μM. This resulted in concentration ratios between Cas12a and crRNA of 1:1, 1:2, 1:3, 1:4, and 1:5, with other component concentrations unchanged. The effects on fluorescence signal intensity under these different concentration ratios were compared, and the optimal ratio was selected for further use.

### 3.6. Optimization of the ssDNA Concentration

The concentrations of ssDNA were tested at 100 nM, 200 nM, 300 nM, 400 nM, and 500 nM, while keeping the concentrations of other components constant. The effects of different ssDNA concentrations on fluorescence signal intensity were observed to determine the optimal reaction concentration.

### 3.7. Sensitivity for the RPA-CRISPR/Cas12a Assay

To assess sensitivity, *H. aduncum* DNA with an initial concentration of 12.7 ng/μL was used, with ddH_2_O serving as a negative control. The DNA was diluted in a 10-fold gradient, resulting in seven concentration levels ranging from 12.7 ng/μL to 12.7 fg/μL. The RPA amplification products were added to the CRISPR/Cas12a system, and the assay was conducted under optimized conditions. The limit of detection (LOD) of the RPA-CRISPR/Cas12a assay was determined and compared with the results obtained from the RPA reaction alone.

### 3.8. Detection of Commercial Samples Using the RPA-CRISPR/Cas12a Assay

A total of 100 fish, including 40 small yellow croakers, 20 mackerels, and 40 hairtails, were purchased from a fish market in Hangzhou. The fish were dissected under a stereomicroscope, and 10 infected individuals from each species were selected to isolate nematodes from their viscera. After washing with saline, 15 nematodes were randomly selected, and their genomic DNA was extracted for detection using the RPA-CRISPR/Cas12a method. The results were compared with and verified by Sanger sequencing of the PCR products for consistency [8]. The RPA-CRISPR/Cas12a reaction system was as follows: RPA reaction conditions and procedures remained as described above, with the optimized CRISPR/Cas12a system comprising Cas12a protein at 250 nM, 10× buffer at 2 μL, crRNA at 500 nM, RNase inhibitor at 0.5 μL, RPA product at 3 μL, ssDNA at 500 nM, and DEPC-treated water to a final volume of 20 μL.

### 3.9. Experimental Data Analysis

Fluorescence intensity was measured using a real-time fluorescence quantitative PCR instrument (Gentier 48E), and statistical analysis was performed using SPSS 23.00. All experiments were repeated three times, and means and standard deviations were calculated using one-way ANOVA. Results are expressed as mean ± standard deviation. Pairwise comparisons were conducted using the *t*-test, and graphs were plotted using Origin 2021. Statistical significance was denoted as follows: *p* < 0.01 indicates an extremely significant difference (**); *p* > 0.05 indicates no significant difference (ns).

## 4. Conclusions

In this study, we designed specific RPA primers and crRNA for the ITS sequence of *H. aduncum*, optimized the Cas12a/crRNA concentration ratio and ssDNA concentration, and developed the RPA-CRISPR/Cas12a fluorescence detection system. This system can specifically detect *H. aduncum* at 37 °C within approximately 45 min, with a minimum DNA detection concentration of 1.27 pg/μL. Results can be directly assessed using a portable fluorescent PCR instrument or UV flashlight, offering easy operation, a short reaction time, and high specificity and sensitivity. This system meets the requirements for practical detection and represents a significant advancement in field detection technology. It is also a valuable addition to the detection and identification of *Anisakis*.

## Figures and Tables

**Figure 1 molecules-29-04789-f001:**
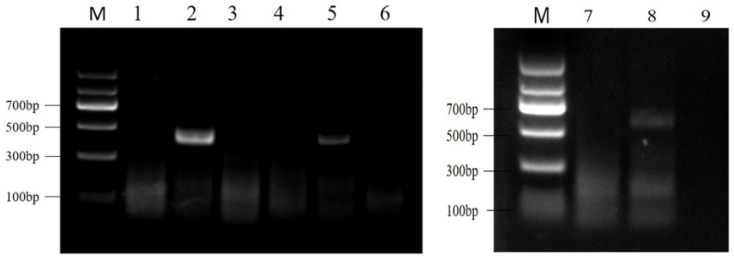
RPA primer optimization results. Lanes 2, 5, and 8 show the amplification results for the genomic primer pairs ADU1, ADU2, and ADU3 of *H. aduncum*; Lanes 1, 4, and 7 are the negative controls using *A. simplex* (*s. s.*); Lanes 3, 6, and 9 are the negative controls using ddH_2_O.

**Figure 2 molecules-29-04789-f002:**
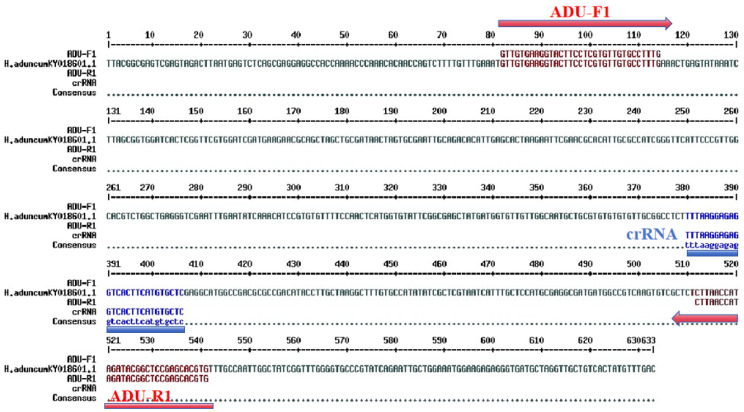
Positions of RPA primers and crRNA target sequences for *H. aduncum*.

**Figure 3 molecules-29-04789-f003:**
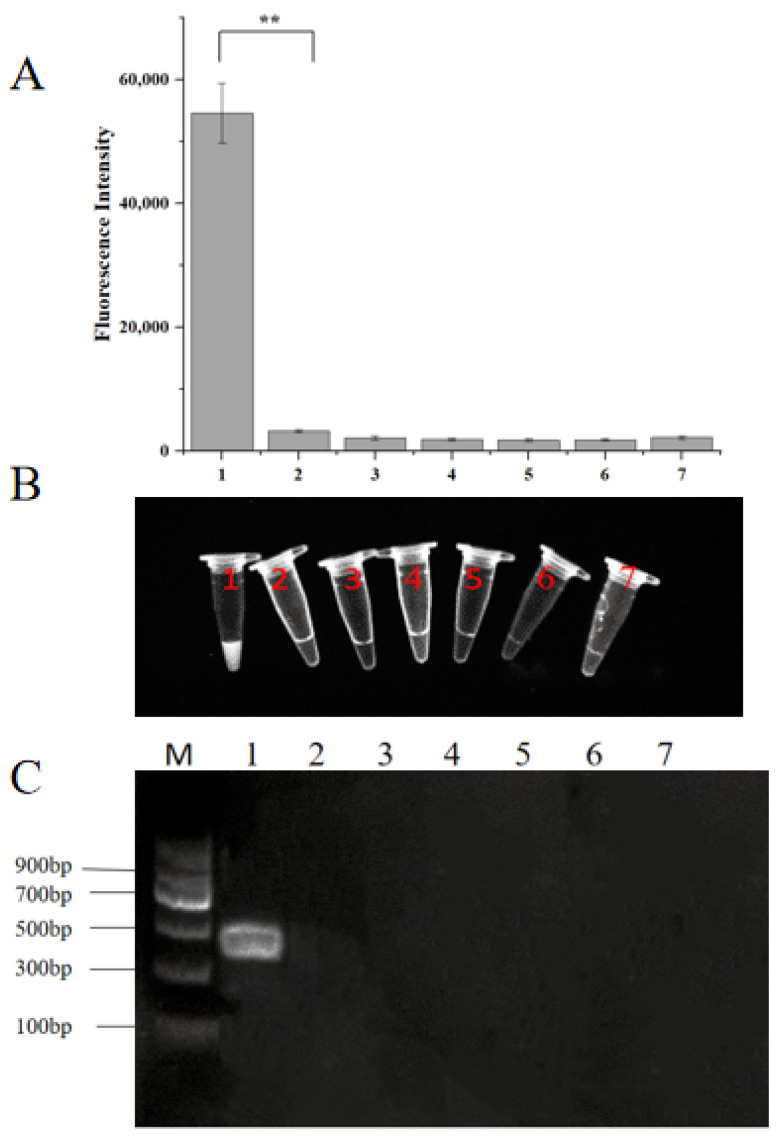
The results for specificity based on the RPA-CRISPR/Cas12a for *H. aduncum*. (**A**) Endpoint fluorescence analysis of the RPA-CRISPR/Cas12a assay. (**B**) Fluorescent color development under UV light of the RPA-CRISPR/Cas12a assay. (**C**) Specificity test of the RPA reaction. Lane 1 represents *H. aduncum*; Lane 2 represents *A. simplex* (*s. s.*); Lane 3 represents *A. typica*; Lane 4 represents *H. sinense*; Lane 5 represents *Contracaecum* spp.; Lane 6 represents the aggregate sample of two fish species meat; Lane 7 represents negative control (ddH_2_O was used as a reaction template). ** represent *p* < 0.01.

**Figure 4 molecules-29-04789-f004:**
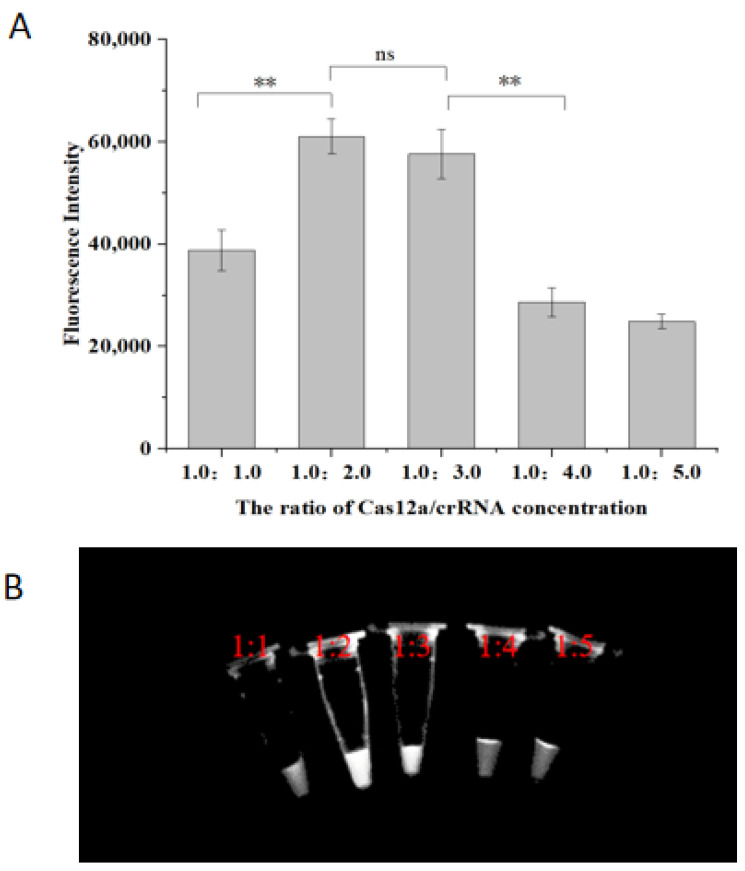
Optimization of Cas12a/crRNA concentration ratio. (**A**) Endpoint fluorescence analysis of the system; (**B**) Fluorescent color development under UV light. ** represent *p* < 0.01; ns represent *p* > 0.05.

**Figure 5 molecules-29-04789-f005:**
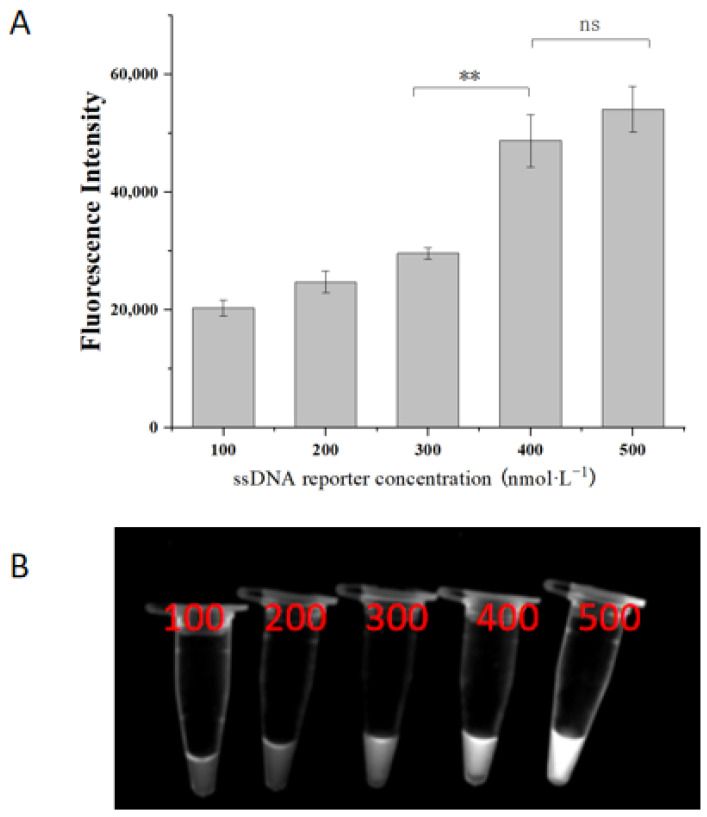
Optimization of ssDNA concentration. (**A**) Endpoint fluorescence analysis of the RPA-CRISPR/Cas12a assay; (**B**) fluorescent color development under UV light. ** represent *p* < 0.01; ns represent *p* > 0.05.

**Figure 6 molecules-29-04789-f006:**
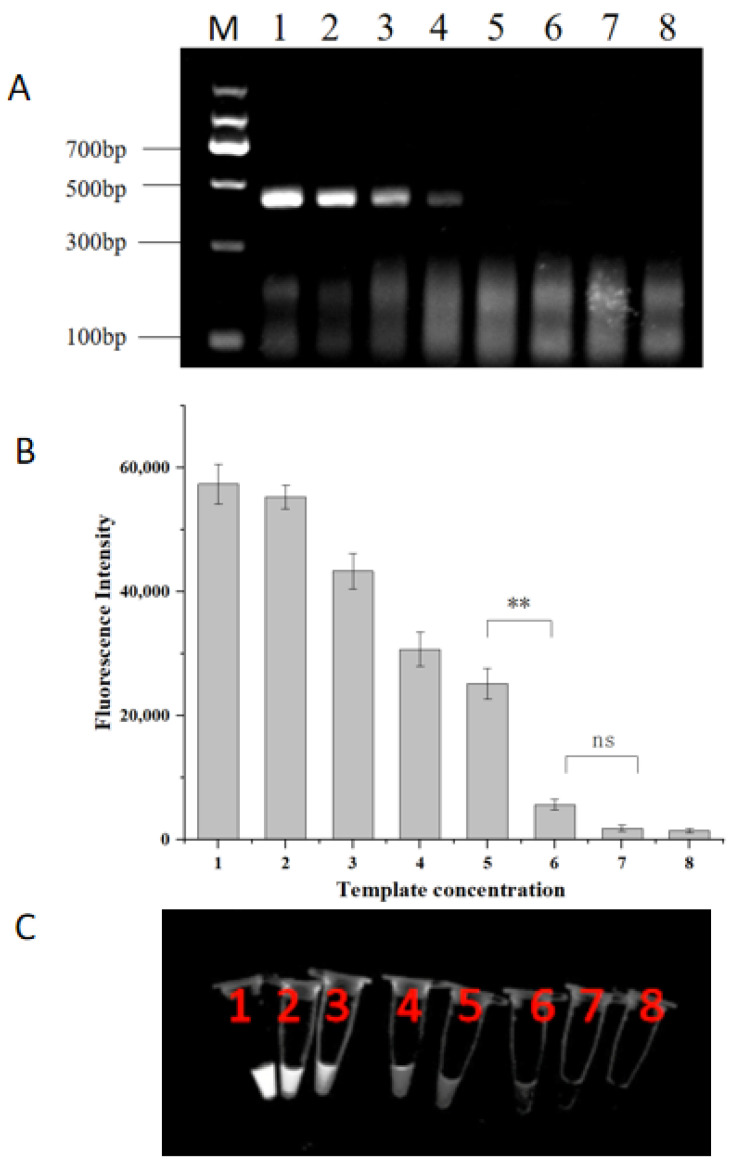
Sensitivity of the RPA-CRISPR/Cas12a system. (**A**) Sensitivity evaluation of the RPA reaction. (**B**) Endpoint fluorescence analysis of the RPA-CRISPR/Cas12a assay. (**C**) Fluorescent color development under UV light of the RPA-CRISPR/Cas12a assay. Lane 1: 12.7 ng/μL; Lane 2: 1.27 ng/μL; Lane 3: 127 pg/μL; Lane 4: 12.7 pg/μL; Lane 5: 1.27 pg/μL; Lane 6: 127 fg/μL; Lane 7: 12.7 fg/μL; Lane 8: ddH_2_O. ** represent *p <* 0.01; ns represent *p >* 0.05.

**Figure 7 molecules-29-04789-f007:**
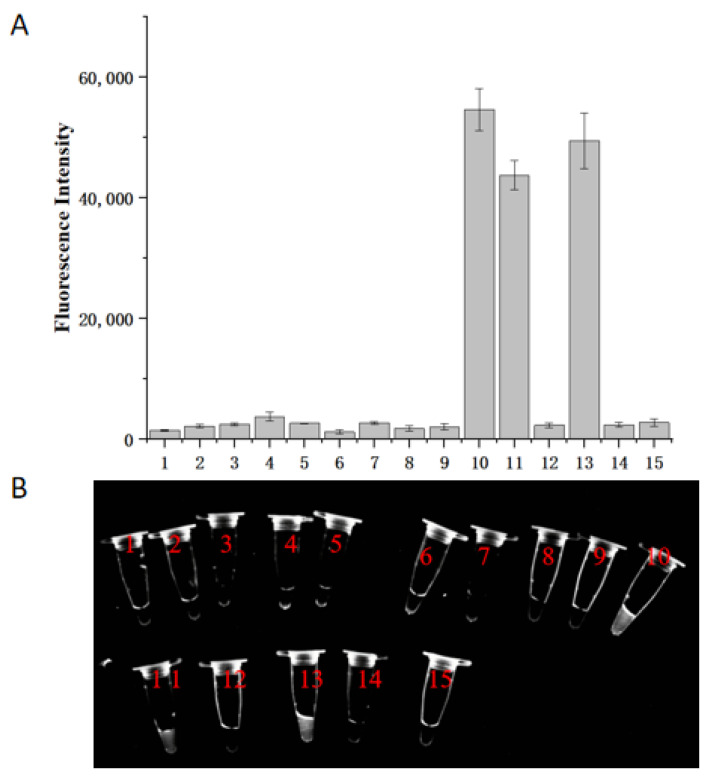
Results of actual sample testing using the RPA-CRISPR/Cas12a assay. (**A**) Endpoint fluorescence analysis of the RPA-CRISPR/Cas12a assay. (**B**) Fluorescent color development under UV light.

**Table 1 molecules-29-04789-t001:** Oligonucleotide sequences.

Primer/Probe		Sequence (5′-3′)	Product Size (bp)
ADU 1	ADU-F1	GTTGTGAAGGTACTTCCTCGTGTTGTGCCTTTG	470 bp
	ADU-R1	CACGTGCTCGGAGCCGTATCTATGGTTAAG	
ADU 2	ADU-F1	GTTGTGAAGGTACTTCCTCGTGTTGTGCCTTTG	420 bp
	ADU-R2	GACACTTGACGGCCATCATCGCCTCGCATGGAG	
ADU 3	ADU-F1	GTTGTGAAGGTACTTCCTCGTGTTGTGCCTTTG	500 bp
	ADU-R3	CTAGCATCACCCTCTCTTCCATTTCCAGCA	
	crRNA	UAAUUUCUAAGUGUAGAUAAGGAGAGGUCA CUUCAUGUGCUC	
	ssDNA	(FAM)-TTATT-(BHQ-1)	

## Data Availability

The data have not been uploaded to a publicly available repository, and the data that support the findings of this study are available upon reasonable request to the corresponding author, Guan.

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
