# Peer review of "Rapid and Ultrasensitive Detection of *H. aduncum* via the RPA-CRISPR/Cas12a Platform"

_molecules, 2024, doi:10.3390/molecules29204789_

Round 1

Reviewer 1 Report

Comments and Suggestions for Authors

The manuscript is devoted to the development of a fast and sensitive method for detecting the presence of nematode H. aduncum DNA in food samples. The authors selected modern and sensitive amplification (RPA) and detection (Cas 12a) methods. This combination really has a very high sensitivity and allows detecting the presence of small amounts of target DNA. The speed of the RPA reaction is also very high. Despite this, the described method still requires the stage of DNA extraction from the samples, which takes up most of the time in any molecular diagnostic methods. As a recommendation for improving the manuscript, I could recommend conducting work on determining the sensitivity of the reaction in copies of the target region. The fact is that using a regular DNA sample with a known concentration as a standard for tenfold dilutions, you cannot say exactly how much genomic DNA is present in your sample. Amplicons obtained by the RPA method can be cloned in the same way as after PCR, using T-A cloning. This way, you will obtain a plasmid containing the target region and will be able to use it as a standard for tenfold dilutions and more accurate determination of sensitivity.

In future work, it would be interesting to test the developed method without the extraction step, using sample swabs or diluted homogenized samples as a matrix for amplification, and compare how much the sensitivity of reaction decreases. Perhaps the method can be accelerated even more by eliminating the DNA extraction step?

The article is written in good English, I could not find any unfortunate phrases or typos. Despite this, there are still several comments:

- Not very high quality of images. In Figure 6C, tubes 6, 7 and 8 are almost invisible. In Figure 7B, the numbers are difficult to read.

- I did not immediately understand what ssDNA means. Perhaps it is better to indicate this component as ssDNA probe?

In conclusion, I would like to add that the manuscript is well written and makes a good impression. It may be of interest to developers of molecular diagnostic tests and parasitologists. In my opinion, the work can be recommended for publication after the deficiencies have been corrected.

Author Response

Comment 1: On the one hand, the method described by the authors still requires the step of DNA extraction, which occupies most of the time for detection and diagnosis, and on the other hand, sensitivity testing using a conventional DNA sample of known concentration as a 10-fold dilution standard may be possible by constructing plasmids for more accurate sensitivity assessment.

Responds 1: We would like to begin by thanking you for recognizing this study and for making a valuable suggestion. We concur with your assertion that DNA extraction is a time-consuming aspect of the complete assay process, which is undoubtedly a pressing issue in this field. In the context of molecular assaying, the combination of the crude DNA extraction technique with molecular assaying may potentially lead to more efficient and rapid assays, particularly in the case of RPA-related techniques, which offer significant advantages in terms of efficiency compared to traditional assays. RPA-related techniques offer significant advantages over traditional detection methods in terms of efficiency, which is a potential avenue for further investigation in subsequent studies. However, the impact of DNA crude extraction on the accuracy of the assay remains a crucial aspect that requires careful consideration. Furthermore, as proposed by the reviewer, a more precise sensitivity evaluation can be conducted by constructing plasmids. This approach was not initially considered, as our primary objective was to enhance the assay's efficiency and sensitivity during the experimental design phase. And many research reports used DNA content as an indicator of sensitivity detection rather than copy number. However, we will contemplate incorporating the methodology suggested by the reviewer in our forthcoming studies.

We extend our gratitude once more for your valuable suggestions.

Comment 2: Not very high quality of images. In Figure 6C, tubes 6, 7 and 8 are almost invisible. In Figure 7B, the numbers are difficult to read.

Responds 2: Thank you for bringing this up, we have revised them.

Comment 3: I did not immediately understand what ssDNA means. Perhaps it is better to indicate this component as ssDNA probe?

Responds 3: As mentioned in line 70 of the text, ssDNA is a type of single-stranded DNA. When the Cas12a protein binds to the crRNA in the system, it has the activity of cleaving ssDNA. It differs from the probe at the point where it does not need to bind to the template, but is waiting to be cut, thus releasing fluorescence. Anyway, thank you very much for your advice.

Reviewer 2 Report

Comments and Suggestions for Authors

The manuscript is well written and easy to read. However, miner comments should be addressed as follows:

1.  Line 61: RPA offers advantages such as rapidity, accuracy, and the ability (please add a reference). The gap between this study and previous investigations should be explained at the end of intro.

2. Line 79: how long did you store genomic DNA?

3. Line 171: please move it to methods. Fig. 3: it should be improved. In the figure 3 legend: you mentioned *, **, ***, however, you illustrate in the figure only **, please correct.

4. Fig. 4: it should be improved. You should state in the figure legends what stars indicate. Please do the same for figure 5-7.

5. The limitations of this study should be described. Conclusion: it is too short.

Author Response

Comment 1: Line 61: RPA offers advantages such as rapidity, accuracy, and the ability (please add a reference). The gap between this study and previous investigations should be explained at the end of intro.

Respond 1: Thank you for pointing this out. We have added references as requested in line 64 ( reference [14, 15])

Comment 2: Line 79: how long did you store genomic DNA?

Respond 2: For this point, the DNA samples we stored were extracted within the range of 1-3 months, and the OD value of DNA was detected before use.

Comment 3: Line 171: please move it to methods. Fig. 3: it should be improved. In the figure 3 legend: you mentioned *, **, ***, however, you illustrate in the figure only **, please correct.

Respond 3: Thank you for your pointing this out it. we have moved it to methods as line 258 and the content related to the legend notes has been uniformly corrected

Comment 4: Fig. 4: it should be improved. You should state in the figure legends what stars indicate. Please do the same for figure 5-7.

Respond 4: Thank you for your advices. As shown in respond 3, we have made corrections in the legend note.

Comment 5: The limitations of this study should be described. Conclusion: it is too short.

Respond 5: Agree. We have added the limitations of this study as lines 241-247, but we consider the conclusions of the aforementioned article to be an accurate and comprehensive summary of our study, and therefore have not included any additional descriptions.

Reviewer 3 Report

Comments and Suggestions for Authors

Reviewer comments:

The manuscript ID molecules-3211117 entitled “Rapid and Ultrasensitive Detection of H. aduncum via the RPA-2 CRISPR/Cas12a Platform” I found this research topic is interesting, developing a faster, more accurate, and simpler detection method for H. aduncum. But I have few concerns related to the research article. I am asking authors to revise the manuscript carefully considering my comments for possible publication in “Molecules”.

Abstract:

Good

Introduction:

• Line No 61: The authors are requested to “expand RPA”.

• Line No 143: The authors are requested to check and correct “(12.7 ng/μL to 12.7 fg/μL)”.

• Line No 171: The sentence is not clear “Twenty and fifteen DNA samples were extracted”

References:

• Figure 1 and Figure 3 gel images are not good; the authors are requested to replace the PCR images if possible.

The submitted manuscript may be acceptable for publication after a minor revision.

Author Response

Comment 1: Line No 61: The authors are requested to “expand RPA”

Respond 1: Recombinase polymerase amplification (RPA) was a relatively novel isothermal amplification technique based on recombination proteins since 2006, and it was widely used in several detection fields. We add it in line 61,

Comment 2: Line No 143: The authors are requested to check and correct “(12.7 ng/μL to 12.7 fg/μL)”.

Respond 2: Thank you for your questions, but as the Fig.6 legend notes show that the concentration include 12.7 ng/μL、 1.27 ng/μL、 127 pg/μL、 12.7 pg/μL、 1.27 pg/μL、127 fg/μL and 12.7 fg/μL. So we think it is right.

Comment 3: Line No 171: The sentence is not clear “Twenty and fifteen DNA samples were extracted”

Respond 3: As for this question, the “Twenty” refers to the laboratory storage samples we mentioned in Method 3.1, and the “Fifteen” refers to the unidentified samples taken from the sampling we mentioned in Method 3.8.

Comment 4: Figure 1 and Figure 3 gel images are not good; the authors are requested to replace the PCR images if possible.

Respond 4: Thank you for your advice. However, in order to ensure the effectiveness of the primer screening process, we do not plan to change Figure 1. In general, this figure indicates the quality of the PCR results and meets the needs of result analysis. Of course, we have processed the electropherogram in Figure 3 to ensure that the results are easier to observe without changing the experimental results.

Reviewer 4 Report

Comments and Suggestions for Authors

Authors have developed and validated a RPA-Cas9 based methodology to detect H. aduncum, one of the causative pathogens of anisakiasis. Even though this technology has been widely used in detecting other organisms, the present work clearly demonstrates the applicability of this technique to detect this parasite in a rapid an sensitive way.

There are, however, some points that authors should address or correct:

1. Did the authors test the applicability of this technique to viscera from infected fish or isolated nematodes instead of purified DNA?

2. Authors should consider to move panels B and C in Figure 3 to supplementary materials. They already showed the DNA amplification in Figure 1, and panel 2 is just a visual representation of panel 1. Same for Figures 4 to 7.

3. Figure 4: X-axis label. Please indicate the ratio versus what.

4. Figure 5, panel A, X-axis: "ssDNA reporter concentration" instead of "The concentration of the ssDNA reporter". Same in Figure 6, panel B (template concentration instead of "the concentration of the template").

Comments on the Quality of English Language

English only needs some minor editions.

Author Response

Comment 1: Did the authors test the applicability of this technique to viscera from infected fish or isolated nematodes instead of purified DNA?

Respond 1: Thanks your question, In fact, we have carried out experiments on artificial contamination of nematode samples mixed in fish meat in the previous RPA-LFD study1. In principle, in the previous study, RPA-LFD and the CRISPR/Cas12 system in this study are both visually detected by the amplification products after RPA reaction, and there is no difference in nature, so this study is based on the consideration of accurate identification of H. aduncum in genus and species. Practical detection tests are carried out by purified DNA from nematodes or mixed DNAs from nematodes and fish.

Comment 2: Authors should consider to move panels B and C in Figure 3 to supplementary materials. They already showed the DNA amplification in Figure 1, and panel 2 is just a visual representation of panel 1. Same for Figures 4 to 7.

Respond 2: Considering that this study aims to develop a method, the process of establishing and optimizing methods should be presented in detail. So we arranged the three observations together to make it easier for the reader to make a visual comparison, and it seems that there is a certain degree of reproducibility, as suggested by the reviewers. However, if this requirement is to be met, the layout and language of the entire article will need to be significantly revised, and we have decided not to move it because it will not affect the integrity and logic of the article. But all in all, the reviewers' suggestions are greatly appreciated, and we will pay special attention to this in future studies.

Comment 3: Figure 4: X-axis label. Please indicate the ratio versus what.

Respond 3: This ratio pair is the ratio of Cas12a protein to crRNA added to the system, we have modified it in new Figure 4, Thank you for pointing this out.

Comment 4: Figure 5, panel A, X-axis: "ssDNA reporter concentration" instead of "The concentration of the ssDNA reporter". Same in Figure 6, panel B (template concentration instead of "the concentration of the template").

Respond 4: Thank you for pointing this out, we have modified it in new Figure 5 and Figure 6

  1. Wang, X.; Xu, T.;  Ding, S.;  Xu, Y.;  Jin, X.; Guan, F., Recombinase polymerase amplification combined with lateral flow dipstick assay for rapid visual detection of A.simplex (s. s.) and A.pegreffii in sea foods. Heliyon 2024, 10 (7), e28943.

Round 2

Reviewer 4 Report

Comments and Suggestions for Authors

Authors have included most of the suggestions except for the ones requested regarding the figures. However, they did explain the reasons why they decided to keep the figures as in the original version, so it is ok with me.